# Feasibility of a Supervised Postpartum Exercise Program and Effects on Maternal Health and Fitness Parameters—Pilot Study

**DOI:** 10.3390/healthcare11202801

**Published:** 2023-10-23

**Authors:** Carla Brites-Lagos, Liliana Ramos, Anna Szumilewicz, Rita Santos-Rocha

**Affiliations:** 1ESDRM, Department of Physical Activity and Health, Sport Sciences School of Rio Maior, Polytechnic Institute of Santarem, 2040-413 Rio Maior, Portugal; 200500015@esdrm.ipsantarem.pt (C.B.-L.); lilianaramos@esdrm.ipsantarem.pt (L.R.); 2CIEQV, Life Quality Research Center, Polytechnic Institute of Santarem, 2040-413 Rio Maior, Portugal; 3Department of Fitness, Faculty of Physical Culture, Gdansk University of Physical Education and Sport, 80-336 Gdansk, Poland; anna.szumilewicz@awf.gda.pl; 4CIPER, Interdisciplinary Centre for the Study of Human Performance, Faculty of Human Kinetics, University of Lisbon, 1499-002 Cruz Quebrada, Portugal

**Keywords:** exercise, physical activity, postpartum, exercise prescription

## Abstract

The postpartum period is marked by profound changes in women at physical, psychological, and physiological levels. Many of these changes persist after four to six weeks postpartum, and most women do not resume their levels of physical activity, which increases the risk of remaining inactive for many years. It is crucial to implement effective programs that promote exercise during the postpartum period. The objective of this study was to test the feasibility and analyze the effects of a structured and supervised postpartum exercise program on maternal health and fitness parameters. To analyze the potential effects of the intervention, the level of physical activity, quality of life, pelvic girdle and low back pain, fatigue, depression, and the level of functional and physical fitness were assessed at baseline, after 8 weeks, and after 16 weeks of intervention. Feedback on the exercise program was collected after the final assessment. The results showed that a structured and supervised postpartum exercise program was feasible and safe and produced positive effects on selected maternal health and fitness parameters. These results will encourage a study protocol with a larger sample in order to prove its effectiveness, improve the guidelines for postpartum exercise, and incorporate this program into a routine healthcare setting.

## 1. Introduction

After delivery and after completing the anatomical and functional changes of pregnancy, the reversal process begins. Many of the physiological and morphological changes of pregnancy persist for four to six weeks after delivery [1]. The puerperium or postpartum period can last up to a year and is marked by a great emotional vulnerability for the woman, as it is a transition period. It implies profound changes at the physical, psychological, and sociological levels. Some physical and psychological health conditions resulting from pregnancy persist after this period [1]. Among the most prevalent issues arising from pregnancy are postpartum weight retention, musculoskeletal complications such as pelvic pain, low back pain, abdominal diastasis, pelvic floor problems, and psychological complications such as postpartum depression [2]. The benefits of exercise for the general population are well known [3]; however, specifically for this period, the benefits include recovery from childbirth, promotion of return to pre-pregnancy weight, reduced risk of developing future chronic conditions of health, improvement of fitness parameters, interactions between mother and baby, and social interactions [4]. In addition to these benefits, recent systematic reviews have shown that exercise in the postpartum period is effective in weight loss [5] and in reducing symptoms of depression [5,6,7,8,9], musculoskeletal disorders [10,11], and fatigue [12].

The U.S. Department of Health and Human Services (USDHHS) [13], the World Health Organization (WHO) [14], and the American College of Sports Medicine (ACSM) [15] issued guidelines on physical activity and sedentary behavior for postpartum women, reinforcing the need for national policies to include and monitor this subpopulation. Specific guidelines for physical activity during the postpartum period are embedded in the documents supported by the American College of Obstetricians and Gynecologists (ACOG) [1], the Brazilian Society of Cardiology (SBC) [16], and Sports Medicine Australia (SMA) [17]. One consensus paper supported by the International Olympic Committee (IOC) addresses exercise and postpartum in recreational and elite athletes [2]. Since these international guidelines refer to the late postpartum and lack specific content regarding the implementation of postnatal exercise programs, an updated textbook chapter about “Exercise Prescription and Adaptations in Early Postpartum” [18] was previously published.

Even so, there is a lack of public health policies related to exercise in both early and late postpartum periods, as well as a scarcity of studies in this area. Moreover, it is important to understand and implement effective strategies that promote exercise during the postpartum period.

Considering the importance of exercise in the postpartum period, the characteristics of women in this period, and the recommendations of the main international organizations, a specific exercise program was developed and validated [19].

The aim of this pilot study was to test the feasibility and analyze the effects of the “Active Mums” postpartum exercise program on maternal health and fitness parameters.

## 2. Materials and Methods

### 2.1. Participants

The sample consisted of Portuguese postpartum women.

Participants were recruited through social networks (Facebook and Instagram), as well as for convenience. The inclusion criteria were postpartum women between 18 and 45 years old, with no medical contraindications for the practice of physical exercise, and understood the Portuguese language. Exclusion criteria were any medical contraindications for physical exercise. During the recruitment stage, it was explained that the exercise program could be performed in person or online, according to their residence.

### 2.2. Equipment and Materials

To implement this exercise-based intervention aimed at improving a comprehensive set of maternal health and fitness parameters, the following equipment and materials were used to support the promotion and planning of the program, to implement the program, and to collect and process data. These tools were chosen because they may comprehensively evaluate outpatient postpartum recovery.

Planning of the program:Informed consent.Participant’s form with sociodemographic and clinical information.Checklist and individual registration sheets.Computer to insert and analyze data.

Collecting and analyzing health, physical activity, and quality of life parameters:Physical Activity Readiness Questionnaire for Everyone (PAR-Q+) [20], adapted to Portuguese [21].International Physical Activity Questionnaire (IPAQ) [22], adapted to Portuguese [23].World Health Organization Quality of Life Questionnaire (WHOQOL-Bref) [24], adapted to Portuguese [25].Pelvic Girdle Questionnaire (PGQ) [26], adapted to Portuguese [27].Roland-Morris Disability Questionnaire (RMDQ) [28], adapted to Portuguese [29].Fatigue Assessment Scale (FAS) [30], adapted to Portuguese [31].Edinburgh Postpartum Depression Scale (EPDS) [32], adapted to Portuguese [33].Blood pressure and resting heart rate, using monitors [34].Body Mass Index (BMI), using a scale and a stadiometer [34].Waist-hip ratio, using a tape measure [34].Body composition, using a TANITA RD 545 Bioimpedance scale [34].

Collecting and analyzing functional and fitness parameters:Postural assessment [34], using a camera.Battery of functional tests [35,36,37], using a mattress, chair, measuring tape, and a stick.Cardiorespiratory fitness, using a treadmill and a heart rate monitor [38,39].Muscular resistance, using a mattress and a chair [40].Flexibility, using a mattress and a measuring tape [41].

Implementation of the exercise program:Sports equipment used in exercise sessions: mattresses, Swiss balls™, elastic bands, free weights, kettlebells, TRX™, softballs, ergometers, pulleys.Computer for online provision of exercise sessions.

Promotion of the exercise program

Support exercise manual [42]Free download guides “Active Pregnancy Guide-Physical activity, nutrition, and sleep” [43] (also available in English [44]) and “Promotion of physical activity and exercise during pregnancy and postpartum. Health professionals guide” [45] (also available in English [46]), and free access videos posted on the YouTube channel “Gravidez Ativa–Active Pregnancy”: https://www.youtube.com/channel/UC0Vyookwc0mcQ5T70imtoNA/playlists (accessed on 21 September 2023).

### 2.3. Intervention Program

The postpartum exercise program consisted of a supervised, personalized training program carried out in-person or online, individually or in small groups of up to four participants. Each participant was invited to attend 3 of the available exercise sessions per week for 16 weeks (48 sessions in total). The exercise program was periodized into 3 weekly sessions of 60 min, referring to 3 mesocycles: adaptation (first two weeks), improvement (six to eight weeks), and maintenance. A typical 60 min session includes low-impact cardiorespiratory training, postural and functional/resistance training (core, lower and upper limbs, back), neuromotor training (balance and coordination), pelvic floor muscle training, and stretching, breathing, and relaxation exercises. However, the exercises are personalized, considering the needs and specificities of each woman, especially taking into account the type of birth, the early postpartum stage (0–6 weeks), and the late postpartum stage.

The intervention program is described in the study “Development and Validation of the Physical Exercise Program “Active Mums” for Postpartum Recovery. Qualitative study with application of the CReDECI-2 Guidelines” [19], and in the exercise manual “Prescription of Physical Exercise in the Postpartum Period” [42].

### 2.4. Tasks, Procedures and Protocols

For the promotion and implementation of the exercise program, the following procedures were followed.

Before the intervention:Preparation and promotion of the educational materials [42,43,44,45,46].Training qualified exercise professionals [47].Preparation of the physical practice space: authorization request to the person responsible for the “Lateral Performance” training space to carry out the program, as well as the use of the respective equipment.Promotion of the exercise program on social networks (Facebook and Instagram).Recruitment of participants by completing an online form.Explanation to the participants about the objective and pertinence of the study and the importance of their collaboration and availability in this research.Informed consent: statement informing the study objectives, information about the assessment sessions, confidentiality, participation and abandonment, damages related to the investigation, exclusion criteria, and disclaimer.Medical clearance for the postpartum exercise program.

Initial assessments at baseline:Initial individual interview to fill out the participant form to obtain sociodemographic and clinical information, as well as potential barriers and preferences regarding the intervention.Measurement of parameters of physical activity, quality of life, health, physical fitness, and functionality (between 6 and 8 weeks postpartum).Measurement of blood pressure and resting heart rate, as well as calculation of reserve and maximum heart rate.Assessment of BMI and body composition.Measurement of waist and hip perimeters, calculation of the waist-hip ratio.Postural assessment, with static observation of the anatomical references in the frontal and sagittal planes, verifying the symmetry in relation to the imaginary midline and photographing the same.Application of some tests from the Battery of Functional Aptitude Tests Dynamic Neuromuscular Stabilization (DNS), namely the seated diaphragm test, intra-abdominal pressure test, quadruped rockforward test, and qualitative recording of the results.Application of some tests from the Battery of Functional Aptitude Tests Functional Movement Screen™ (FMS™), such as shoulder mobility, active straight leg raise, deep squat, rotary stability, and qualitative recording of results and scores.Assessment of cardiorespiratory fitness using the Rockport One-Mile Fitness Walking Test and recording time, heart rate, and calculation of maximum oxygen consumption (VO_2max_).Assessment of muscular endurance, counting the maximum number of arm extensions (push-ups) and applying the “Chair Stand Test,” registering the maximum number of repetitions in 30 s.Assessment of flexibility, with the “V-sit and reach test.”Intermediate (after 8 weeks) and final assessment (after another 8 weeks):Assessment of parameters of physical activity, quality of life, health, physical fitness, and functionality.

After the final assessment:A form was sent to participants to collect feedback on the level of satisfaction with the exercise program, containing the following questions inspired by Haakstad et al. [48]:
Level of satisfaction with the program.Level of satisfaction with the instructor.Do you consider that exercise in a group environment was/would be more motivating than if it were individual?In what parameter (s) did you feel improvements in terms of your physical fitness?Have you changed your physical activity levels?Do you feel more energy for daily activities and less stress?Would you recommend this program to a friend?Would you participate in the program again after another pregnancy?Would you like to leave other comments?


### 2.5. Data Processing

Data were recorded in Excel, where descriptive statistics were performed.

### 2.6. Ethical Considerations

A group of healthy postpartum women was invited to participate in the pilot intervention free of charge. Participants were informed about the purpose and nature of the study, the potential benefits for future programs, participation requirements, and their right to withdraw from the study. They were free to provide feedback or not, without any consequences, and their feedback was anonymous. All women were informed and agreed to participate in the program, in physical and functional assessments, and in assessments by questionnaires. An informed consent was signed prior to participation. The educational materials produced by the research team were made available to the participants free of charge.

All exercise sessions and assessments were conducted by a qualified exercise physiologist. All clinical appointments were conducted by a gynecologist. The study was conducted in accordance with the Helsinki Declaration. This study is part of the study protocol that has been approved by the Ethics Committee of the Polytechnic Institute of Santarém, Portugal (approval number 9-2021-ESDRM).

## 3. Results

### 3.1. Participants Characterization

The sample consisted of 11 Portuguese postpartum women. At the beginning of the program, 8 of them were between 6 to 8 weeks postpartum, and 3 of them were between 9 and 12 weeks postpartum, aged between 24 and 37 years old (mean age of 31 years). Two participants had a cesarean delivery by medical decision, while the remaining nine had vaginal deliveries, with two using suction cups. Eight women were primiparous, while the other three were multiparous. Nine women were residents in Leiria, Portugal, and carried out the program in person at the Lateral Performance training studio in Leiria, while the rest were residents in Alcochete and in Pombal, attending the online program but making themselves available to do in-person assessments. They were all married or in a relationship. Regarding the education of the participants, nine had higher education, and two had completed high school.

Except for one participant, all women in the sample practiced regular exercise before pregnancy (e.g., bodybuilding, group classes, handball, personalized training). During pregnancy, only seven women exercised (e.g., walking, personalized training for pregnancy, group classes, Pilates), and before starting the program, only one declared that she practiced exercises at home. As for the pre-gestational Body Mass Index (BMI), the values varied between 20.8 and 34.4 kg/m^2^ (mean of 25.2 kg/m^2^), with three women being overweight and one obese. Self-reported maximal gestational weight gain ranged from 8 to 25 kg (mean of 20.2 kg).

### 3.2. Level of Physical Activity, Quality of Life, and Other Health Parameters

Table 1 contains the results obtained at the three collection times.

#### 3.2.1. Level of Physical Activity

Before starting the program, the mean of habitual physical activity of the group was 559 min/week. After eight weeks, it increased to 810 min/week, and at the end of the program, it increased to 1127 min/week. This trend was expected since it includes the time spent on the exercise program. The section in which the most time was spent in a normal/usual week was “Housework, House Maintenance, and Caring for Family,” with a weekly mean of 327 min before the start of the program, rising to 676 min in the second evaluation and ending with 504 min.

When analyzing the sections separately, it was observed that “Job-related Physical Activity” is the section to which the participants did not dedicate any time since they were all on maternity leave, a situation that remained similar until the end of the program.

In the “Transportation Physical Activity” section, it was observed that, initially, four participants did not use physical activity as a means of transport to move from one place to another, while the rest used walking as a means of transport. There was a similar situation in the remaining evaluative moments. Despite this, there was an increase in the weekly travel time from 127 to 193, ending at 180 min.

In the physical activities carried out in the “Housework, House Maintenance, and Caring for Family” section, it was found that only one participant did not perform such activities, with a minimum duration of ten continuous minutes, while the rest were dedicated to domestic tasks with moderate intensity, before starting the program. This situation was not maintained, with all participants starting to carry out these moderate activities and increasing their weekly means, as already mentioned.

Regarding the section that considers “Recreation, Sport, and Leisure-time Physical Activity,” it is noteworthy that only three of the participants did not perform these activities weekly with a minimum duration of 10 continuous minutes before the exercise program, totaling a mean time weekly of 106 min. Walking was the physical exercise adopted during their free time, with only one participant who revealed that she performed moderate and vigorous activities. In the second evaluation, and considering that they participated in the exercise program, the same section started to present 289 min/week of physical activity. At the end of the program, the participants stated that they had performed a mean of 443 min of physical activity per week, thus quadrupling their initial value.

The “Time Spent Sitting” section is related to the time that the participants remain seated during the week and at the weekend. All participants in this study spent a lot of time in the sitting position during the week and on the weekend before starting the program, but during the weekend, the mean was 324 min/day, while during the week, the mean was 345 min/day. In the next evaluation, the mean time for each one decreased to 257 and 197 min/day, respectively. The downward trend continued, and after the program, the participants reported a mean of 165 min/day on the weekend and 156 min/day during the week. Thus, on weekends, the time spent sitting was greater than on weekdays.

#### 3.2.2. Quality of Life

In this parameter, there was no change between domains through the WHOQOL-Bref analysis. Before starting the program, the highest score in terms of quality of life, from 0 to 100, was observed in the social relationships domain (81.98 ± 10.42), followed by the environment domain (81.3 ± 11.6), the physical domain (79.4 ± 6.45), and the psychological domain (75.4 ± 12.4). The general mean was 79.54, indicating that the participants have a good quality of life. In the following evaluation, the domain with the highest score was the psychological (82.4 ± 10.1), followed by the physical domain (81.2 ± 9.15), the environment domain (80.8 ± 12.7), and the social relationships domain (78.1 ± 9.99), inverting the disposition of the first evaluation and achieving an overall mean of 80.6. At the end of the program, it was again the psychological domain that obtained the highest score (86 ± 9.74), followed this time by the social relationships domain (82.66 ± 12.27), the environment domain (80.28 ± 11.97), and the physical domain (79.72 ± 9.74), indicating an improvement in almost all domains compared to the first assessment. Thus, the mean was 82.17 (±9.71).

In the initial assessment, when asked about their quality of life, most participants (91%) reported that it was “good.” There was a similar situation in the following evaluative moments. As for satisfaction with their health, most participants (91%) reported being satisfied, a trend that continued until the end of the program.

#### 3.2.3. Pelvic Pain and Low Back Pain

Regarding pelvic pain, with the application of the PGQ, a large difference was noted among women, ranging from zero (four participants) to 18 points (out of a possible 75), with a mean of 5.8 (±7). None of the participants reported a critical situation. There was a significant improvement in the intermediate evaluation, reaching a mean of 1.5 (±3.2), in which six participants no longer had pain. At the end of the intervention, pelvic pain became almost non-existent, having had a mean of 0.4 (±0.7), with only two participants reporting some type of pain.

As for low back pain, with the application of the RMDQ, no differences were reported during and after the exercise intervention. The initial mean score before starting the exercise program was only 1.2 (±1.7) out of 24, with five women not reporting any type of pain in the daily activities described, and the maximum score obtained was 5. Similar scores were reported in the following evaluations.

#### 3.2.4. Fatigue

In the FAS questionnaire, each item is evaluated on a five-point Likert scale, where 1 corresponds to “never,” and 5 corresponds to “always,” ranging from 10 points (less fatigue) to 50 points (more fatigue). Before starting the program, the mean was 22.4 (±5.4). In the following evaluations, the mean slightly decreased to 21.6 (±5.4). These FAS scores indicated that women were not very fatigued in any of the evaluation moments.

#### 3.2.5. Depression

Before starting the program, in accordance with the EDPS, only two participants reached a score equal to or greater than 12, which is considered a risk factor for the onset of postpartum depression, according to Cox et al. [32]. The other women had scores between two and eight, considered within the normal range, thus indicating a lower risk for the onset of postpartum depression. The baseline mean was 6.3 (±3.7). In the following assessments, the mean decreased to 5 (±3); therefore, none of the participants presented a risk of postpartum depression since the maximum score was eight.

### 3.3. Functional and Health-Related Components of Physical Fitness

#### 3.3.1. Body Composition

There was no change in the mean values of postpartum body weight between the first, second, and last evaluations. Of the participants in this study, five were overweight, while in subsequent assessments, there were only four, according to the BMI classification. Thus, their mean BMI at baseline was 26.8 kg/m^2^ (±4.7), decreasing to 26.5 kg/m^2^ (±5.0) after 8 weeks and ending with 26.1 kg/m^2^ (±5.2).

Regarding other anthropometric variables, such as waist and hip perimeters and their respective ratios, all participants showed differences in the different assessments, losing a mean of 4.1 cm in the waist from the initial assessment to the second and losing a mean of 1.6 cm from the second to the last assessment. In the hip perimeters, the differences between the initial and the following evaluations were smaller, with a mean loss of 3.3 cm, and the same trend continued for the following evaluation, with a mean loss of 3 cm. In this way, the waist-hip ratio did not suffer major changes, with only one person having a ratio greater than 0.85 and maintaining it until the end of the program.

Regarding the body composition variables assessed by bioimpedance, it is important to highlight the percentage of fat mass and muscle mass. The percentage of fat mass registered favorable differences in the different evaluations, showing mean values of 34.7% (±5.6) at the initial moment, changing to 33.5% (±6.3) in the intermediate evaluation and ending with 28.3% (±10.3). Muscle mass, on the other hand, registered an increasing trend in the different evaluations, initially having a mean of 42.2% (±5.6), 42.6% (±5.8) in the second evaluation, and ending with 43.1% (±6.2).

#### 3.3.2. Cardiorespiratory Fitness

Regarding blood pressure, all participants showed normal values [15]. However, we noticed an apparent decrease in mean values throughout the program, both in Systolic Blood Pressure (SBP) and Diastolic Blood Pressure (DBP).

As for mean rest heart rate, there was a slight decrease from the first to the second assessment and maintenance between the second and the last assessment.

The Rockport 1-mile fitness walking test was used to assess aerobic fitness. The predicted VO_2max_ values in the initial evaluation had a mean of 29.4 (±7.1) mL/kg/min, a value considered “average” for this age group, according to the ACSM [15]. In the second evaluation, the mean value of predicted VO_2max_ increased to 31.9 (±4.5) mL/kg/min, considered “good,” and in the last evaluation, the mean increased to 37 (±6.2) mL/kg/min, reaching the threshold of “excellent”.

#### 3.3.3. Muscular Endurance and Flexibility

The participants performed arm extensions until they were unable to perform more executions or until they lost body alignment. The mean of push-ups before starting the program was 16 (±9.9) repetitions. In the following evaluation, the mean was 19 (±8.3) repetitions, ending, in the final evaluation, with 25 (±11.4) repetitions. The “Chair Stand Test” was performed for 30 s, obtaining means of 14 (±2.4), 16 (±1.9) and 19 (±3.4) repetitions, respectively. In the “V-Sit and Reach” test, the participants performed a mean of 43 (±7.0) cm, 45 (±7.1) cm, and 47 (±7.7) cm in the first, second, and third evaluations, respectively.

These scores are considered “normal” for this age group.

#### 3.3.4. Postural Assessment

A qualitative postural assessment was performed. In the postural analysis, in the anterior view, the puerperal women presented a good alignment of the anatomical references in the initial evaluation. Regarding the position of the visually evaluated body segments, the majority of the participants had the head aligned, knees and iliac crests aligned, and feet supinated. Most of them also had a smaller earlobe-shoulder distance on the side of their dominant hand. In the following evaluations, the postural pattern was maintained.

In the sagittal plane, in the initial evaluation, most of the participants had a forward head, cervical lordosis, protruding shoulders, hyperkyphosis, lumbar hyperlordosis, hip joint in anteversion, knees in hyperextension, and misaligned external malleoli. In this plane, several differences were noticed in the following evaluations, and most participants (91%) ended the program with the head aligned, the neck normal, the shoulders normal, without hyperkyphosis, lumbar spine and iliac crest normal, coxo-femoral joint normal, and knees and malleolus external aligned.

#### 3.3.5. Functional Assessment

A qualitative DNS functional assessment was performed. In the “Seated Diaphragm test,” the women had the ability to expand the abdominal wall and perform a symmetrical activation without elevating the ribs and shoulders and without losing the verticality of the spine. A similar situation was observed in the following evaluations.

Regarding the “Intra-abdominal Pressure test,” in the initial evaluation, most participants (64%) presented a hyperextension of the lumbar area, as well as a weak activation of the abdominal wall with bulging and a slight diastasis of the rectus abdominis. In the following evaluation, only three women had the same condition. At the end of the program, all participants performed correct activation of the abdominal wall and stabilization of the lower back.

Regarding the “Quadruped Rockforward test,” in the initial evaluation, most of the participants (82%) presented a cervical hyperextension, bringing the head to reclining, an unequal load of the palms of the hands, a scapular elevation, and anterior pelvic inclination. In subsequent evaluations, all participants were able to activate their abdominal, back, diaphragm, and pelvic muscles, straightening their spine.

Qualitative and quantitative FMS functional assessment was performed. Regarding the “Shoulder mobility test,” in the initial evaluation, the wrists of most women were at a mean of 8.5 cm apart on the right and 12.8 cm apart on the left, changing to 6.3 and 9.4, respectively, and then to 6.3 and 8.3. They did not report any pain.

In the “Deep Squat test,” six women (55%) were able to perform the movement without a plate, with the hips parallel, the tibia and trunk parallel, the knees aligned over the toes, symmetrical, without noticeable lumbar flexion, without the feet turning externally, without taking the heels off the ground and without pain; two performed the movement with pain. In the following evaluation, only one participant could not perform the movement correctly, and none of them had pain. A similar situation was observed in the final evaluation.

In the “Straight leg raise test,” only one woman was unable to perform the correct movement. All the others were able to perform the movement, keeping the opposite hip neutral, the toes pointed upwards, the opposite knee in contact with the board, and without pain. A similar situation is in the following evaluations.

In the rotary stability test, most women performed the exercise with the spine parallel to the plate, hips parallel to the floor, knees and elbows aligned with the plate, the support ankle dorsiflexed, touching the elbow to the knee, and without pain, but only contralateral. At the end of the program, only one participant managed to do homolateral.

### 3.4. Satisfaction with the Program

As for the level of satisfaction with the program, 91% of the participants reported being very satisfied, while one reported being satisfied. All participants reported being very satisfied with the exercise professional. In total, 45% agreed that it is more motivating to exercise in a small group than in an individual setting, while 45% strongly agreed and one did not agree. All participants reported an improvement in their physical fitness, namely in terms of strength (81.8%), cardiorespiratory fitness (54.6%), flexibility (36.4%), posture (81.8%), body composition (72.7%), and balance and coordination (45.5%). 91% of women reported increased levels of physical activity, while one reported that she was equally active. All participants felt more energy for daily activities and less stress, reporting that they would recommend the training program and confirming their participation in a possible future postpartum period. There were no substantial differences in the opinions and level of satisfaction between online and in-person participants.

## 4. Discussion

The aim of this pilot study was to test the feasibility and analyze the effects of the “Active Mums” supervised postpartum exercise program on selected maternal health and fitness parameters.

At the psychological level, it is known that postpartum depression is experienced by approximately 20% of women; however, up to 50% of women experience high levels of depressive symptoms during this period [49]. When analyzing the depression results, we found that, initially, only two women scored levels that could be related to postpartum depression, a situation that, over the course of the program, disappeared. These results are in accordance with Carter et al. [8], who demonstrated that exercise is effective in reducing symptoms of depression in postpartum women.

The systematic review of Wilson et al. [50] concluded that there is a strong correlation between fatigue and depressive symptoms among women in the first two years after childbirth. However, our study did not confirm this outcome, as apparently, the levels of fatigue were higher than those on the depression scale, in addition to the fact that there were no significant decreases throughout the program, as we found on the scale of depression.

When analyzing the pain level of the participants through the RMDQ and the PGQ, we verified that in the first one, no differences were observed between the different evaluations and that the mean found was 1 in relation to the scores of functional capacities in the RMDQ, which can range up to 24. This situation led us to question the validity of the RMDQ in the immediate postpartum period. In the PGQ, although the participants did not appear to have a high degree of disability, some women reported pelvic pain in the initial evaluation, which decreased throughout the program until becoming scarce at the end of it. These results are in line with existing literature that has reported a positive association between exercise and decreased pain in the lumbar and pelvic region [50,51,52].

In the current literature, there is a correlation between postpartum depression and quality of life [53,54]. The results of the WHOQOL-Bref showed that most of the women participating in this study had a good quality of life right from the start and that it continued to improve throughout the program, as happened with the symptoms of depression, apparently supporting this correlation. Our results also support the findings of Yang and Chen [55] regarding the impact of an exercise program on fatigue.

Regarding the physical parameters, with regard to changes in body mass and BMI, we found that there was a decrease in values, with almost half of the participants returning to pre-pregnancy values. These results were not expected, according to Meyers and Hong [56], who concluded that exercise had no significant effect on weight loss in the short or long term in breastfeeding women. The same study [56] also reported findings about exercise and body composition, with exercise reducing body fat and preserving fat-free mass.

In this pilot study, there was a loss of fat mass of about 1.2% after 8 weeks and about 6.4% after 16 weeks of intervention, and an increase in muscle mass of 0.4% in the second evaluation and 0.9% after 16 weeks of intervention, which may explain the slight loss of body weight throughout the program. The outcome of the four participants who were still overweight at the end of the program was related to gestational weight gain, which was higher than the recommendations based on pre-pregnancy BMI, according to the Institute of Medicine. However, Dalfra et al. [57] suggested that appropriate gestational weight gain should be personalized considering the three obesity classes. As for the other anthropometric measures used, such as waist circumference, these showed a decrease by the end of the program. The waist-hip ratio shown before the start of the program was within healthy values, except for one participant, and this ratio remained unchanged until the end of the program.

All the changes already mentioned can be potentially explained by the exercise program implemented and the consequent increase in physical activity. Despite the fact that postpartum women are still recommended not to perform physical activity during the puerperal period, as confirmed by the participants of the program, the results of the level of physical activity using IPAQ were positive. Thus, the level of physical activity before starting the program was very low, limited to “Housework, House Maintenance, and Caring for Family.” Regarding the mean time spent per week on physical activity, there was an exponential increase from that initial period until the end of the program in the sections “Housework, House Maintenance, and Caring for Family” and in the section “Recreation, Sport, and Leisure-time Physical Activity.” “Sitting time” is an indicator of a sedentary lifestyle in IPAQ, having decreased throughout the program, both during the week and at the weekend. Thus, our participants progressively met the recommendations of the WHO [14] and ACOG [1], which include: postpartum women should start by doing small amounts of physical activity and gradually increase frequency, intensity, and duration over time; performing at least 150 min of moderate-intensity aerobic physical activity during the week, as well as incorporating a variety of aerobic and muscle-strengthening activities; adding gentle stretching may also be beneficial; and limiting the amount of time being sedentary. Therefore, our program seemed to positively influence the physical activity habits of these participants in their daily lives.

In terms of musculoskeletal parameters, postpartum postural analysis proves to be extremely important, as postural changes are frequent among puerperal women, both due to the gait biomechanical compensations during pregnancy [58,59,60] and potentially due to the tensions and overloads generated with the baby care and breastfeeding. In our study, before starting the program, all mothers presented a similar postural profile, with changes in pelvic tilt and lumbar and thoracic curvatures being the most noticeable, in line with the study by Gilleard et al. [61]. Biviá-Roig et al. [62], when analyzing the lumbar curvature in pregnant, non-pregnant, and postpartum women, hypothesized that it is the muscular responses, and not the curvatures, that are altered by pregnancy. This finding may encourage further research on whether the increase in tension in the pelvic muscles is due to the progressive displacement of the center of gravity during pregnancy and the abdominal weakness due to the diastasis of the rectus abdominis.

Postural compensations resulting from the dominance of the upper limb were also noted, such as a rise in the shoulder on that side in the frontal plane. During the program, strengthening and/or stretching exercises were used and combined, which made it possible to notice great postural differences at the end of the program. Before starting the program, such postural and muscular weaknesses were noticed when the DNS and FMS™ functional tests were applied. At the DNS level, before starting the program, the greatest fragility was noted in the abdominal muscles, while in the FMS™, weaknesses and asymmetries in movement were noted. After a qualitative analysis of the same, we concluded that there was a very positive progression in the functionality of the participants, mainly between the first and the second evaluations. Flexibility exercises can also improve postural stability, especially when combined with resistance exercises. In this line, these types of exercise were inserted into the program, and the respective assessments were made, leading to positive outcomes in flexibility and muscular resistance. This observation may encourage further research regarding the effectiveness of neuromotor exercise and the inclusion of this type of exercise in the guidelines for exercise prescriptions for postpartum women.

The functional state of the respiratory, cardiovascular, and musculoskeletal systems depends on cardiorespiratory fitness, which is related to health [15]. The cardiorespiratory fitness of the participants obtained very positive results, progressing from an “average” condition in the initial assessment to an almost “excellent” condition at the end of the exercise program, according to the ACSM’s VO_2max_ classification [15]. In short, it was demonstrated that the fitness parameters showed an improvement, which is consistent with the available research regarding the importance of exercise programs for the postpartum period.

As for satisfaction, it was evident that the program met or exceeded the participants’ expectations, emphasizing the importance of group sessions, the improvement of physical fitness, and the satisfaction with the exercise professional. The levels of satisfaction highlight the importance of the supervision of pre and postnatal exercise programs by qualified exercise professionals, preferably involved in multidisciplinary teams [63]. Supervision of exercise programs is recommended to ensure proper technique, provide confidence, and ensure the progression of appropriate levels of intensity and complexity. The exercise professional should provide regular feedback, positive reinforcement, and behavioral strategies to enhance adherence. The exercise professional should also provide the safest possible training and testing environment, prevent exercise-related emergencies, and be familiar with the safety and emergency procedures available at the fitness setting where the exercise program is delivered.

Regarding the feasibility of the postpartum exercise program both in person and online, it was demonstrated to be safe, tailored, motivating, and a contribution to improving the quality of life, functional capacity, and physical fitness of postpartum women, as shown by either qualitative or quantitative tools such as questionnaires and physical fitness tests performed. These results will encourage a study protocol with a larger sample in order to prove its effectiveness and incorporate this program into a routine healthcare setting.

The main limitation of this study is the small number of participants, preventing extrapolating conclusions, and the non-use of inferential statistics to support the observed trends. Moreover, it is recognized that there are various methodological and ethical constraints when working with this population, as well as a lack of validated tools. As for strengths, we highlight the real context approach used, that is, the implementation of an exercise intervention, combining the advantages of a program for a specific population with the effectiveness of controlled exercise and a progressive stimulus adapted to the level of physical fitness of each participant, guaranteeing their adherence. Due to the number of variables that are involved, this pilot test involves a complexity that can provide guidelines for the preparation of an intervention study protocol with a larger sample.

## 5. Conclusions

This pilot study allowed us to test a tailored supervised postpartum exercise program performed for 16 weeks. This postpartum exercise program was feasible, safe, and motivating and contributed to improving selected parameters of quality of life, physical activity, health, functional capacity, and physical fitness of postpartum women. These results provided guidance to develop a study protocol with a larger sample in order to prove its effectiveness, improve the guidelines for postpartum exercise, and incorporate this program into a routine healthcare setting.

## Figures and Tables

**Table 1 healthcare-11-02801-t001:** Results obtained at the three collection times.

	1 (Baseline) Before the Intervention		2 (Intermediate) After 8 Weeks of Intervention		3 (Final) After 16 Weeks of Intervention	
	Mean	Standard Deviation	Mean	Standard Deviation	Mean	Standard Deviation
IPAQ (min/wk)	559	±520	810	±1171	1127	±669
WHOQOL-BREF (0–100)	79.54	±8.42	80.63	±9.59	82.17	±9.71
PGQ (0–75)	5.8	±7	1.5	±3.2	0.4	±0.7
RMDQ (0–24)	1.2	±1.7	1.4	±1.7	1	±1.4
FAS (10–50)	22.4	±5.4	21.6	±5.4	21.6	±5.4
EPDS (0–30)	6.3	±3.7	5	±3	5	±3
Weight (Kg)	68.95	±13.83	68.4	±14.88	67.1	±15.18
BMI (Kg/m^2^)	26.8	±4.67	26.5	±5.02	26.1	±5.17
Waist-hip Ratio	0.8	±0.05	0.8	±0.06	0.79	±0.06
Fat Mass (%)	34.7	±5.59	33.5	±6.3	28.3	±10.3
Muscular Mass (Kg)	42.18	±5.6	42.55	±5.84	43.08	±6.17
SBP (mmHg)	107.33	±9.90	100	±8.51	97.78	±9.56
DBP (mmHg)	71.33	±4.95	68.44	±4.33	68.11	±6.27
Rest HR (beats/min)	76	±16.16	74.89	±14.59	74.89	±11.34
VO_2max_ (mL/kg/min)	29.39	±7.08	31.93	±4.53	37	±6.24
Push-ups (number)	16	±9.88	19	±8.25	25	±11.41
“Chair Stand Test” (number)	14	±2.38	16	±1.85	19	±3.38
“V-Sit and Reach” (cm)	43	±7.04	45	±7.09	47	±7.73

## Data Availability

The data presented in this study are available on request from the corresponding author. E-BOOK EXERCISE PRESCRIPTION IN THE POSTPARTUM PERIOD: [Portuguese; ISBN: 978-989-8768-49-0] Available at: https://www.researchgate.net/publication/372866386_PRESCRICAO_DO_EXERCICIO_NO_PERIODO_POS-PARTO_Exercise_Prescription_in_the_Postpartum_Portuguese (accessed on 21 September 2023). E-BOOK ACTIVE PREGNANCY GUIDE: [Portuguese; ISBN: 978-989-8768-27-8] Available at: https://www.researchgate.net/publication/340315748_GUIA_da_GRAVIDEZ_ATIVA_-_Atividade_Fisica_Exercicio_Fisico_Desporto_e_Saude_na_Gravidez_e_Pos-Parto_Active_Pregnancy_Guide_Physical_Activity_Exercise_Sport_and_Health_in_Pregnancy_and_Post-partum_Port (accessed on 21 September 2023); [Spanish; ISBN: 978-989-8768-50-6] Available at: https://www.researchgate.net/publication/372907216_Guia_Embarazo_Activo_-_Actividad_Fisica_Nutricion_y_Sueno_Active_Pregnancy_Guide_Spanish (accessed on 21 September 2023); [English; ISBN: 978-989-8768-50-6] Available at: https://www.researchgate.net/publication/370874994_ACTIVE_PREGNANCY_GUIDE_-Physical_activity_nutrition_and_sleep (accessed on 21 September 2023). E-BOOK PROMOTION OF PHYSICAL ACTIVITY AND EXERCISE DURING PREGNANCY AND POSTPARTUM. HEALTH PROFESSIONALS’ GUIDE: [Portuguese; ISBN: 978-989-8768-36-0] Available at: https://www.researchgate.net/publication/364808045_PROMOCAO_DA_ATIVIDADE_FISICA_E_DO_EXERCICIO_DURANTE_A_GRAVIDEZ_E_O_POS-PARTO_Guia_para_Profissionais_de_Saude (accessed on 21 September 2023); [English; ISBN: 978-989-8768-42-1] Available at: https://www.researchgate.net/publication/364806085_PROMOTION_OF_PHYSICAL_ACTIVITY_AND_EXERCISE_DURING_PREGNANCY_AND_POSTPARTUM_-_Health_Professionals’_Guide (accessed on 21 September 2023).

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
