# Peer review of "Feasibility of a Supervised Postpartum Exercise Program and Effects on Maternal Health and Fitness Parameters—Pilot Study"

_healthcare, 2023, doi:10.3390/healthcare11202801_

Round 1

Reviewer 1 Report

Thank you for the opportunity to read this paper. Very nice work.

There are some points to consider.

Material and methods:

Sample characterization: the first and the third paragraph relates to results, so please delete this information from methods and add it at results. Add a line explaining that the exercise could be performed in person or online, according to their residence.

Explain how and when did you achieve the demographic data and when and how did you measure the weight, waist-hip Ratio, muscular mass, rest HR, blood pressure, etc.

Instead of using point by point to explain the methods and the materials used please write them in a way that any reader could carry out the same study. As you have it, is quite difficult to know all the procedures and protocols that you followed. Also explain why you use each questionnaire.

How many clinical appointments and exercise session were estimated? When were estimated the collection appointments? All this information should be given in the methods.

Ethical considerations:

the N should not be given in this section.

Results

All this section could be shorter by using a table with the results. Then don’t repeat all the numbers that are in the tables, just comment the most important issues.

I miss the analysis of the statistical meaning of the differences. Although the very small numbers, you could verify if the changes between the check points are or not significant….

Explain if you are counting with the intervention or not to provide the minutes of habitual physical activity.

Please use the past or the present but be consistent through the manuscript.

Discussion

Is too long. Please try to resume it.

Conclusions

Is the first time in the hole manuscript that you describe the exercise (3 times per week) …please explain it before.

A revision by a native English speaker is needed

Reviewer 2 Report

This topic is important. Readers may find it of interest. It is recognised that there are various methodological/ ethical constraints when working with this population - these can be acknowledged in the write up. In terms of the write up, the format of this study in the methods section should better reflect relevant information only. The results and statistical analysis needs a complete overhaul to be able to make any credible scientific assumptions. 

I would like to see some more references to support some of the statement claims made in the first paragraph of the introduction (Lines 29-34)

In terms of this first paragraph of the introduction, it does not necessarily reflect the population the paper goes on to discuss. I was expecting to read a paper about women in this first 6 weeks post delivery. When does the ‘early post partum’ period end?

Can physical exercise be renamed as physical activity throughout?

Gestational weight gain- this must be a self reported measure? If so, can you report this. Also, was this the weight gained at the point of birth, end pregnancy, after birth, or at the time the study started – e.g 8 or more weeks post partum?

Up to 8 weeks post partum? – be more specific, how long post partum were the shorter?

Tasks procedures and protocols – write as a paragraph with appropriate detail only. Same with equipment list

2.3 – despite the intervention being described elsewhere, a brief description including duration and frequency is required to contextualise this paper.

2.5 data processing – this is completely inadequate as a description of statistics. Were any statistical tests actually conducted?

Results section – All results are presented in the table so do not need to be repeated word for word in text. There is no mention of any statistical tests at all – therefore what do the results mean? These descriptive percentages are exactly that – descriptive. They do not tell us anything of value from a scientific perspective unfortunately.

Discussion: ‘When analyzing the depression results, we found that, initially, only two women were likely to develop postpartum depression, a situation that, over the course of the program, disappeared.’ – This claim is completely misleading and unfounded. No statistics validate it, and the likelihood of PND reduces over time which could easily account for any reductions.

Similar issues for all discussion points at the moment. No claims of reduction can be made without statistical support. 

Mostly very good. A few areas which could be written more clearly.
